# Employment Quality and Mental and Self-Reported Health Inequities among Latinx Housecleaners: The Safe and Just Cleaners Study

**DOI:** 10.3390/ijerph192315973

**Published:** 2022-11-30

**Authors:** Sherry Baron, Isabel Cuervo, Gary Winkel, Deysi Flores, Ana Gonzalez, Homero Harari

**Affiliations:** 1Barry Commoner Center for Health and the Environment, Queens College, City University of New York, Queens, New York, NY 11367, USA; 2Population Health Science and Policy Department, Icahn School of Medicine at Mount Sinai, New York, NY 10029, USA; 3Make the Road New York, Brooklyn, NY 11237, USA; 4Selikoff Centers for Occupational Health, Institute for Exposomic Research, Icahn School of Medicine at Mount Sinai, New York, NY 10029, USA

**Keywords:** precarious employment, employment quality, housecleaners, community-based participatory research, workplace hazards, health inequities, mental health

## Abstract

Precarious employment, such as housecleaning, is one important structural contributor to health inequities. We used an employment quality (EQ) framework to characterize the impact of employment conditions on mental and self-reported ill-health among Latinx housecleaners in the New York City metropolitan area. Using a community-based participatory research approach, we collected cross-sectional survey data from 402 housecleaners between August 2019 and February 2020 to characterize housecleaners’ EQ and its association with depression, perceived stress, and self-reported health. We also measured work-related irritant eye, skin, and respiratory symptoms, which have been shown in previous research to be associated with housecleaners’ exposure to chemical components of cleaning products. Our housecleaner cohort was largely female and immigrant and most had worked at least five years. Survey items capturing the EQ dimensions of unbalanced interpersonal relations, low material resources, and violations of workers’ rights were associated with increased odds of depression, perceived stress, and self-reported ill-health. Work-related irritant eye, skin, and respiratory symptoms were also independently associated with mental and self-reported ill-health and some of the effects of EQ on health were potentially partially mediated through their association with work-related irritant symptoms. Findings can inform directions for community-based educational and policy initiatives to improve housecleaners’ employment quality.

## 1. Background

The changing nature of the economy, compounded by the economic impacts of the COVID-19 pandemic, has increased the number of workers in work arrangements that are less likely to offer long term employment stability and guaranteed wages and benefits [1,2,3]. The resulting economic shifts associated with these conditions have refocused the attention of public health researchers to think more holistically about the work environment by considering not only specific job hazards but also the quality of the employment relationship [4]. This approach has been operationalized in recent epidemiologic studies through the development of a multidimensional employment quality (EQ) framework [5,6,7]. A growing body of both cross-sectional and longitudinal studies, primarily in Europe and the Unites States, have demonstrated associations between measured components of EQ and physical and mental health [5,6,8,9,10,11,12,13,14,15].

The development and validation of EQ measurement tools is an ongoing process [7], though there is consensus regarding the dimensions useful for characterizing EQ [5,6]. These include: (1) the stability of the employment arrangement; (2) the level of material resources (or earnings); (3) worker rights to social protections, such as unemployment compensation, health coverage, paid leave, and nondiscriminatory employment practices; (4) healthy work hours that are predictable and without overwork or involuntary part-time employment; (5) balanced interpersonal power relations with employers, clients or coworkers free of intimidation and abuse; (6) access to training opportunities that increase skills and employability; and (7) institutional empowerment of workers through protection by labor unions or other worker advocacy institutions.

Health studies applying an EQ framework have demonstrated the greatest health risk for those employed in jobs with the poorest EQ and for whom the term “precarious employment” is commonly used [16]. Additionally, labor patterns in the United States show that women, African Americans, Latinxs, and immigrants are disproportionately employed in precarious jobs, underscoring the potential role for work as a structural factor in contributing to racial and ethnic health inequalities [6]. The pathways researchers have proposed for how EQ is tied to health include: (1) disproportionate exposure to workplace hazards; (2) low wages and decreased social protections that contribute to other structural determinants of ill-health, such as poor housing and food insecurity; and (3) the direct effect of poor EQ on workers’ ability to control life circumstances and protect their health in the workplace and in the community [6].

Applying a holistic framework for examining associations between work and health that considers the role of workplace hazards and EQ together also provides opportunities to identify more comprehensive solutions to reduce the role of work in ill-health and improve health equity [4]. EQ measures have most often been applied to datasets in the United States and Europe that include workers in diverse occupations to facilitate creation of categories or typologies of EQ in epidemiologic studies [5,9,10,13,14,15]. However, even in studies examining a single group of workers, the EQ framework can be useful for identifying specific intervention priorities. We applied this use of the EQ framework in our study, Safe and Just Cleaners/Limpieza Digna y Segura. 

Safe and Just Cleaners is a mixed methods community-based participatory research (CBPR) study exploring the role of chemical exposures from household cleaning products on the health and well-being of Spanish speaking Latinx housecleaners in the New York City (NYC) metropolitan area [17]. Housecleaning work is precarious employment with low EQ since housecleaners frequently earn low wages, have informal work arrangements with high job instability, and often lack access to minimum workplace rights, such as health care and paid sick leave [18,19,20]. Housecleaners’ working conditions also routinely expose them to workplace hazards from chemical components of cleaning products known to cause irritant effects, such as dermatitis, asthma, reduced pulmonary function, and eye irritation [21,22,23,24,25]. 

We report on our analysis of data from a cross-sectional survey collected by the Safe and Just Cleaners study between August 2019 and February 2020 that characterized dimensions of housecleaners’ EQ and the association of those EQ measures with ill-health. We hypothesized that housecleaners’ low EQ was a cause of housecleaners’ ill-health and lower well-being. We also hypothesized that one potential pathway for how EQ affects health is by leading to higher levels of work-related irritant symptoms as workers trade off safer work practices to work quickly and avoid conflicts with their clients, especially given the inherent insecurity of their jobs and earnings. 

Our study posed the following specific research questions: (1) using an EQ framework, what are the employment conditions of Latinx housecleaners in the NYC area; (2) is poorer EQ a potential cause of poorer mental and self-reported health among housecleaners (Figure 1 Pathway A); and (3) if EQ is associated with poorer health and well-being among housecleaners, is the pathway of causation direct or is it partially mediated through the impact of poor EQ on housecleaners’ experiencing work-related irritant symptoms (Figure 1 Pathway B). 

## 2. Materials and Methods

Our overall study used a CBPR approach that emphasized the following elements of community involvement: power and control, responsibility and ownership, participation, and influence [26]. This academic-community partnership arose out of a previous collaboration with our community partner, Make the Road New York (MRNY), to create occupational health training for Latinx construction laborers and cleanup workers following a natural disaster in New York City [27]. To further develop our mutual interest in workplace exposures to cleaning chemicals and to expand our research to include more women workers, together our community-academic partnership proposed and carried out this study of household cleaners. Many of the dimensions of employment quality included in our study were initially raised, developed, and prioritized by our community partner. Our overall project activities attempted to balance research and action for the mutual benefit of all partners, another key CBPR principle.

### 2.1. Survey Item Development Strategy

Our survey was designed to characterize cleaners’ EQ and household cleaning products use patterns. This information, in turn, contributed to our quantitative assessment of housecleaners’ exposures to chemicals in household cleaning products and our development of educational and policy suggestions to improve housecleaner employment conditions and safer cleaning practices. The findings related to housecleaners’ EQ are reported here and data related to product use patterns and exposure levels will be reported separately. As input for the development of the survey instrument, we held seven focus groups, five within NYC and two in suburban communities, to capture housecleaners’ experiences working in urban apartments and suburban homes. The 52 focus group participants were recruited through our community partner’s networks. During these focus groups, we collected information related to the types of cleaning products and practices used and about housecleaners’ experiences with dimensions of EQ. 

To develop EQ-related survey items, we drew on previous surveys [14,15,28,29], but tailored the questions to cleaner-specific employment characteristics, based on our focus group reports and previous research [22,30]. Final item selection was accomplished by consensus among all project partners with the dual goals of collecting data specific to understanding cleaning product use patterns while also prioritizing the collection of EQ-related data to support our project’s educational and policy-oriented actions led by our community partner. Items related to two of the EQ dimensions were not prioritized—access to training and institutional (union) empowerment—since we had already determined these are almost universally unavailable to housecleaners. We still prioritized these EQ dimensions in our community-facing programs, but we focused our survey questions on capturing data on the five other dimensions of EQ. Figure 1 shows our hypothesized pathways for the relationships between these five EQ dimensions and health both through a direct effect of EQ on health and an indirect effect of EQ on cleaning practices and irritant work-related symptoms. 

### 2.2. Survey Items Used to Measure EQ Dimensions

For the first EQ dimension, employment instability, we included survey items related to housecleaners’ employment arrangement and their length of time working as a housecleaning in the United States. To characterize work arrangements, we classified housecleaners into one of the following arrangements: working only as a solo self-employed worker, working as a self-employed person but together with another cleaner, working at least some of the time for a cleaning agency, or working in some other arrangement such as a cleaning day laborer or with a worker-run cooperative. The number of years working as a housecleaner was measured with a 5-point response score, with 0 being less than 1 year and each additional point representing an additional 5 years of housecleaning work experience in the United States up to 15 or more years.

The second dimension, work hours, was characterized by two survey items: the total weekly work hours and the number of different clients served per week. We included both these aspects as survey items because moving among different clients might create additional scheduling and time burdens. 

For the third dimension, material resources, we included three survey items. The first was reported monthly earnings from housecleaning, measured using a 5-point response score with 0 being less than $500 dollars per month and each additional point adding an additional $500 earning per month up to $1500 or more. The remaining 2 survey items captured the housecleaners perceived financial insecurity based on their monthly earnings and their insecurity about having enough clients to maintain those earnings, both of which were measured on Likert-like response options. 

For the fourth dimension, workers’ rights, we included survey items related to whether they experienced unfair wages, defined as being paid below the minimum wage or not being paid their full wage (wage theft). We also included items assessing their access to benefits such as sick leave and health insurance. Finally, we included a series of questions related to experiences of seven types of discrimination at work based on: gender, age, race, ethnicity, being an immigrant, the language they speak, and their sexual orientation [31]. For the discrimination questions we created a continuous variable by summing the number of different types of discrimination they experienced at work. 

To characterize the fifth dimension, interpersonal power relations with the employer, we focused exclusively on the cleaner-client relationship since our focus group data pointed to that as the primary power dynamic housecleaners experienced. In developing these survey items, we drew on the well-developed job content construct used to characterize the psychosocial work environment and job-related stress [32], since in our focus groups housecleaners emphasized the importance of time demands, limitations on housecleaners’ power to make decisions about cleaning product use and practices, and the degree to which their clients were supportive or abusive. We included nine survey items, two items related to time demands, three items related to the level of control by clients over cleaning practices and four items related to the level of social support or abuse from their clients. These survey items also used Likert response options, except for the item related to experiences of verbal abuse from their client which used a simple yes/no response. 

### 2.3. Work-Related Irritant Respiratory, Skin, and Eye Symptoms

To assess workers experiences of work-related irritant health effects, we created survey items related to symptoms of work-related respiratory, skin, and eye irritation, consistent with previous studies showing increases in these types of symptoms among housecleaners [22,24,25]. For symptoms of work-related skin irritation, we used a survey question included in the National Health Interview Survey 2010 Occupational Health Supplement [33]. For respiratory symptoms, we included questions asking whether a health care provider ever said they had asthma [34] and an item asking about symptoms of night-time shortness of breath. The latter, which has been found to be highly specific for bronchial hyperreactivity, is especially useful for populations, such as ours, with limited access to health insurance and for whom work-related respiratory conditions may be underdiagnosed [35]. We considered a respondent to have respiratory symptoms if they reported either asthma or night-time shortness of breath. To assess the work-relatedness, we asked if the respiratory symptoms improved when away from work [36]. Given frequent reports of eye irritation among housecleaners [25], we developed a survey item with a 4-point severity response score based on the degree of interference with housecleaners’ ability to work, with 0 being no eye irritation and 3 being irritation so bad they need to leave the room where they are cleaning. 

### 2.4. Mental and Self-Reported Health Outcome Measures

To evaluate mental health, we used the Center for Epidemiologic Studies Depression 10-item Scale (CES-D-10) [37] and Cohen’s Perceived Stress 10-item Scale (PSS) [38] using the Spanish versions of both surveys used in the Hispanic Community Health Study [39]. For overall health, we used the single-item 5-point self-reported health scale which has been shown to be highly predictive of longer-term morbidity and mortality [40]. 

### 2.5. Individual-Level Covariates

We included survey items related to demographic characteristics and other individual factors that might influence health including educational attainment, time in the United States, English language comfort, and family composition including whether they were their family’s primary wage earner. 

We initially developed or identified survey items in English and then bilingual team members translated them into Spanish, except when using previously validated translations of existing measures. When selecting items in English, the study team considered the items’ cultural or linguistic translatability for a NYC-based Spanish speaking population. In addition, we performed cognitive and pilot testing to assure that survey items were interpreted as intended and made edits directly to the Spanish version, when indicated, prior to fielding the survey [41]. 

### 2.6. Participant Recruitment, Survey Administration, and Participation Rate

Our fundamental approach to outreach was to work through community-based organizations known and trusted by our target Latinx communities and to capture the experiences of housecleaners working in urban apartments and suburban homes. We recruited participants in four of five boroughs in NYC and two suburban neighborhoods in Westchester County, New York with a focus on neighborhoods with the highest Latinx density based on American Community Survey data [42]. We used a variety of outreach methods including: (1) neighborhood street outreach; (2) announcements at our community partners’ ongoing meetings and classes; (3) fliers at institutions serving the Latinx community including elementary schools, foreign consulates’ service center offices, and employment centers, and (4) referrals from participants. We collected screening eligibility information from potential participants to assure that they: (1) worked as a housecleaner within the past 2 months, which we defined as work involving cleaning bathrooms and kitchens in apartments or houses on a regular basis at least once per week and (2) had worked as a housecleaner for at least 6 months in the United States. 

The survey was administered by fluent Spanish speakers either at the offices of a community partner or in publicly available quiet spaces, such as the public library, and responses were directly entered into a tablet or computer. Study data were collected and managed using the REDCap electronic data capture tool hosted at the Icahn School of Medicine at Mount Sinai [43]. Visual aids, such as for questions with Likert-type response options, were used. On average, the survey lasted between 60 and 75 min. Our community partners provided each participant a package of referral materials to housing, health, and other services at the end of the survey and participants received a 30-dollar gift card. We rescheduled interviews a minimum of three times before a potential participant was considered a nonrespondent.

We collected screening eligibility information from 925 potential participants and 327 were eliminated—of those eliminated, 301 (92%) had not worked within the previous 2 months and 26 (8%) had worked in housecleaning for less than 6 months. The remaining 598 participants were invited to participate in the study and 419 (70%) completed the survey. Of the 419 participants, 17 left the survey session after completing less than half of the survey and are not included in this analysis, leaving 402 housecleaners included in our analysis. Information on the participation rate by recruitment approach may help others recruiting similar hard to reach worker populations. We found that street outreach had the lowest participation rate (62%) and recruitment through our community partner’s classes and social programs had the highest participation rate (75%). 

### 2.7. Data Analysis

The analytic strategy proceeded in stages. Initially, survey items focusing on individual characteristics and the different aspects of EQ were assessed descriptively (research question 1). Next, we used principal components analysis (PCA) to determine whether the questionnaire items designed to capture the same EQ dimension could be grouped into a single variable. The criteria for determining if individual EQ items could be grouped included having an eigenvalue greater than one and component item loadings greater than 0.40. Since the EQ items themselves were measured using ordinal scaling, the inter-item measure that formed the basis of the PCA analysis was the polychoric inter-item correlation. A similar PCA process was used to assess whether the survey items related to work-related irritant symptoms (respiratory, skin and eye) could be combined into a single variable. For items meeting the PCA criteria to be combined, we generated a single variable by summing the scores for the individual items.

Once the combined EQ and irritant symptom variables were developed, we examined statistical associations between the EQ variables and the irritant symptom variable using multiple linear regression. In this modeling, we included individual characteristics such as age, education level, and English proficiency as well as the EQ variables. This analysis was performed as the first step of a mediation analysis to test whether some of the effects of EQ on mental and self-reported health were mediated through the impact of low EQ on housecleaners’ work practices generating higher hazardous exposures and irritant symptoms, (research question 3 and pathway B in Figure 1). The measure of association in the linear regression model was based on the significance levels associated with each of the EQ independent variables. 

Finally, we created multivariable logistic regression models including our EQ-related variables as the independent variables and our health outcomes (depression, perceived stress, and self-reported health) as the dependent variables, while adjusting the model for any significant individual level covariates (pathway A in Figure 1 and research question 2). For the logistic regression outcome measures, we dichotomized our health measures using the recommended threshold cut point for the CESD-10 (≥10) [44] and for the PSS we used a cut point (≥14) suggested for health screenings [45] and consistent with the range of the mean score found in the Hispanic Community Health Study [46]. For self-reported health, we used those reporting poor or fair health (compared to those reporting good, very good, or excellent health). In calculating the CESD-10 and PSS scores, we included all participants who had non missing data for at least nine of the 10 questions and assigned a response value for the 10th missing value based on the average response score on the other nine questions. Statistical modeling used backward elimination of non-significant variables and the adjusted R-square measure as programmed for SAS [47]. The Pearson Chi-Square value was used to test the Goodness of Fit of our models. Statistical significance was based on the 95% confidence intervals.

Once we selected the best logistic regression model (model 1), we developed a second model by adding the irritant symptom variable to model 1. Model 2 was used for two objectives: (A) to determine if the irritant symptom variable impacted the mental and self-reported health outcomes; and (B) to determine if the inclusion of the irritant symptom variable in the equation for each of the three outcomes mediated the effects of any of the EQ variables on health outcomes (pathway B in Figure 1 and research question 3). If the addition of the irritant symptom variable in any of the equations for stress, depression, and/or self-reported health impacted the statistical significance of any of the EQ variables, which would be indicated by a reduction in the statistical significance of the odds ratio (OR) for the EQ variable under consideration, and that EQ variable was also associated with the irritant symptom score in the linear regression model, we concluded there was evidence for at least partial mediation [48].

Participants with missing information on a particular characteristic or outcome were only excluded from analyses involving that characteristic or outcome. Analyses were done using SAS 9.4; Version 15.1.

## 3. Results

### 3.1. Sample Characteristics

Our sample was almost entirely (99%) female. The sample’s average age was 44; standard deviation (SD) 10.3; all were foreign-born but had lived in the United States on average 15 years (SD 9.0); 122 (30%) have only completed a primary school education; and only 14% reported feeling comfortable with spoken English. Many (44%) were the primary wage earner for their family. Demographically, our sample was very similar to the estimated 343,527 housecleaners in the United States in 2019, based on an analysis of microdata from the US Census Current Population Survey, and was especially like the 61.5% of cleaners nationally who are Hispanic or Latino [20].

### 3.2. Employment Quality

Descriptive findings of the EQ-related survey items are summarized in Table 1 and Table 2. Over half of our sample had worked more than five years in housecleaning in the United States and, as is typical of the industry, most (74%), were self-employed and worked alone or together with another housecleaner while 16% reported working for a cleaning agency, at least sometimes. Most housecleaners (74%) reported being either very or extremely worried about having enough clients to earn the money they needed and 20% reported their earnings were insufficient to meet basic needs. Monthly earnings from cleaning were low with 55% earning less than $1000 per month. Work hours were generally part-time and they cleaned for an average of three clients per week, mirroring findings from other national housecleaner surveys [49].

Lack of access to health insurance was very common (49%). Only 15% reported having access to paid sick leave and 48% reported not being able to take sick leave, whether paid or unpaid, because of fear of employer retaliation. While the average wage rate was $15 per hour, instances of unfair pay occurred commonly with 44% reporting pay below the minimum wage and 20% reporting not being paid the amount the client promised (wage theft) in the last year. A third of housecleaners (34%) reported some experiences of discrimination at work and the most common forms of reported discrimination experienced at work were due to being an immigrant, their ethnicity, or the language they speak.

Housecleaners also reported unbalanced interpersonal power relations with clients, including clients having control over how housecleaners’ work was done (68% reported that clients always/almost always chose the cleaning products and 33% of cleaners sometimes/never decided the order of cleaning tasks) and, to a lesser extent, they experienced time pressures (20% always/almost always felt pressure to work quickly and 20% always/almost always did not follow safety and health precautions to finish more quickly). Housecleaners also reported unsupportive client behavior (29% reported clients sometimes or never valued or praised their work and 20% sometimes or never provided them with the tools they need) and 18% reported verbal abuse from clients. Many housecleaners (57%) reported experiencing communication barriers, at least sometimes, with clients due to language spoken (Table 2).

### 3.3. Work-Related Irritant Symptoms

Our survey found the following work-related conditions: 16% reported respiratory symptoms that improved when away from work, which included either an asthma diagnosis by a health care provider (5%) or night-time shortness of breath (13%); 27% reported skin rash that improved when away from work; and 84% reported some level of eye irritation at work (mild irritation 11%, moderate 30%, severe 33%) (Table 3).

### 3.4. Depression, Perceived Stress and Self-Reported Health

Using cut points for the CES-D-10 (≥10 on a 30-point scale) and Cohen’s PSS (≥ 14 on a 40-point scale), we found that 24% of housecleaners could be classified as depressed, while 44% experienced moderate or severe stress. We found that 29% of respondents reported fair or poor versus good/very good/excellent self-reported health (Table 3).

### 3.5. Principal Component Analysis (PCA)

Using PCA, the survey questions designed to capture the same dimension of EQ were assessed to determine whether they could be combined into a single variable. This process allowed us to reduce the number of items in three of the EQ dimensions: interpersonal power relations; material resources and workers’ rights (Table 4).

We also used PCA to assess whether the irritant symptom questions (respiratory, skin and eye) could be combined into a single irritant symptom variable, with the presence of skin or respiratory symptoms each adding 1 point and the eye symptoms adding up to 3 points (mild = 1, moderate = 2, severe = 3). The descriptive statistics and the component loadings for both the irritant symptom and EQ items that were combined, as well as the name for the combined variable, are shown in Table 4.

### 3.6. Associations between EQ Variables and Work-Related Irritant Symptoms

As a preliminary step in our mediation analysis, we examined the associations between EQ variables and the work-related irritant symptoms score using multiple linear regression analysis. We found that variables from the dimensions of unbalanced power relations with clients (time pressures and unsupportive or abusive clients), low material resources (job-related financial insecurity), and inadequate workers’ rights (number of experiences of discrimination at work and lack of health insurance) were all associated with more irritant symptoms (Table 5).

### 3.7. Logistic Regression Modeling for Associations between EQ, Irritant Symptoms, and Mental Health and Self-Reported Health

Table 6 summarizes the results of our separate logistic regression models for depression, perceived stress, and self-reported health. Model 1 shows results only including statistically significant individual and EQ-related variables. In model 2, we added the measure for work-related irritant symptoms to model 1 to test both the contribution of work-related irritant health effects on overall mental and self-reported health and to examine the potential for some of the associations between EQ and health to be mediated through the association between EQ and irritant symptoms.

In model 1, we found that measures capturing the EQ dimensions of unbalanced interpersonal relations with clients, low material resources, and violations of workers’ rights were associated with all three health outcomes. In addition, we found a dose response relationship between more years of work as a cleaner and poorer self-reported health. Working for an agency compared to being self-employed was associated with lower levels of depression, though this finding was based on a small sample of agency workers and requires further evaluation.

The individual demographic variables important in the models include associations between discomfort with spoken English and poorer self-reported health, which mirrors our finding of the association of language barriers in client communication (a measure of poor interpersonal power relations with clients) and higher perceived stress. The other individual variable, cleaners who reported being the primary family wage earner, had increased odds for being depressed which was consistent with the higher odds for depression for those with higher job-related financial insecurity (a measure of low material resources).

In model 2, reporting more work-related irritant symptoms was associated with both mental and self-reported health outcomes with odds ranging between 1.37 and 1.54 independent of the EQ variables. In this second model we also assessed the potential for partial mediation of the effects of low EQ on health through the effect of poor EQ on housecleaners’ work-related irritant symptoms (pathway B in Figure 1). We examined how the addition of the work-related irritant symptom variable changed the ORs for the EQ variables that were found to be associated with a higher work-related irritant symptom score on our linear regression analysis (Table 5). Regarding unbalanced power relations, we found the OR for unsupportive or abusive clients was no longer statistically significant in the model for perceived stress, though the lower 95% CI for the unadjusted OR was close to 1.0 and the 95% CIs were wide. In the model for depression, the OR for unsupportive or abusive clients was also reduced though it remained statistically significant. Additionally, in the model for self-reported poor or fair health, the point estimate for the OR for time pressures was similarly reduced but remained statistically significant. Regarding the dimension of workers’ rights, in the self-reported health model, experiences of discrimination at work were no longer statistically significant in model 2, but, like the finding for unsupportive and abusive clients in the stress model, the lower 95% CI for the unadjusted OR was close to 1.0 and 95% CIs were wide. Finally, for the EQ dimension for low material resources, the point estimate for job-related financial insecurity was reduced in model 2 for both the depression and perceived stress models but both remained statistically significant in the model.

## 4. Discussion

In our study of Latinx housecleaners in the NYC metropolitan area, we applied an EQ framework (Figure 1) to explore how EQ in combination with work-related irritant symptoms affects housecleaners’ health. Measured dimensions of EQ that were most strongly associated with mental and self-reported ill-health in our study include low material resources (job-related financial insecurity and inadequate earnings), workers’ rights abuses (unfair pay and experiences of discrimination at work), and unbalanced interpersonal power relations with clients (unsupportive and abusive clients, client-initiated time pressures, and barriers to client communication due to language). Work-related irritant symptoms were common in our study, which is consistent with previous research among housecleaners suggesting that exposures to chemical components of household cleaning products can cause respiratory, dermatologic, and ocular irritant health effects [22,24,25]. We found housecleaners’ reports of irritant symptoms were also associated with poorer mental and self-reported health independent of EQ. Though EQ and irritant work-related symptoms were mostly independently associated with mental and self-reported health, we also found some evidence of at least partial mediation of interpersonal power relations with clients and violations of workers’ rights through work-related irritant symptoms. This could potentially be the result of poorer EQ resulting in higher exposures to workplace hazards, as workers trade off safer work practices to work quickly and avoid conflicts with their clients, especially given the inherent insecurity of their jobs and earnings.

Our study provides support for all three hypothesized pathways through which diminished EQ could affect health and contribute to health inequities. First, EQ seems to impact exposures to workplace hazards, as we found that several components of housecleaners’ EQ are associated with higher levels of irritant symptoms potentially due, at least in part, to work practices that generate exposure to hazardous cleaning chemicals. Second, housecleaners’ low material rewards likely create other structural determinants of ill-health such as poor housing and food insecurity, though we did not directly measure these factors. Third, our finding that client-related time pressures are associated with poorer self-reported health may reflect the challenges housecleaners confront in controlling life circumstances within and outside the workplace given their dual roles as workers and primary family wage earners and caregivers [50,51,52].

Underlying all these pathways lies the fundamental structural obstacles that immigrants face in finding formal employment that offers better EQ, despite many working 10 or more years as a housecleaner. Housecleaners’ experiences of discrimination at work, which included discrimination due to being an immigrant and the language they speak, was statistically associated both with poorer self-reported health and work-related irritant symptoms, providing further support for additional studies that examine work-related exposures as one of the structural pathways through which discrimination causes health inequalities for immigrants [20]. Taken together, these findings support recent recommendations for a more holistic approach when defining occupational health exposures to consider both EQ as well as hazardous workplace exposures, especially among precariously employed workers such as housecleaners [4].

We found that housecleaners’ mental health outcomes were associated both with housecleaners’ experiences of poorer EQ and their reports of irritant work-related symptoms, consistent with recent calls to increase research that explores the effects of structural determinants of health, including employment conditions, on mental as well as on physical health [53]. Our finding of associations between housecleaners reports of working for unsupportive and abusive clients and depression is consistent with a large body of previous research looking at the relationship between the psychosocial work environment and workers’ mental health [54]. In addition, we found that other aspects of EQ that have been less well studied were also associated with mental health outcomes including financial insecurity, the housecleaner being the primary wage earner of their family and reports of workers’ rights abuses. Finally, though several studies have documented how housecleaners’ work exposures are associated with respiratory, skin and eye irritation, the impact of these exposures on workers’ mental health have been less well studied and deserve attention. These findings point to the importance of including a multidimension EQ framework in future studies aiming to understand the complex relationship between work, stress, and health [55].

Important to our study and guided by our CBPR approach, we prioritized generating timely actionable results and leadership development opportunities for study participants and our use of the EQ framework has supported this process. Our study findings led us to prioritize development of programs and support for policies that address key aspects of housecleaners’ precarious EQ [56]. The project created the Super Cleaners group, a safe space organized by our community partner, Make the Road New York (MRNY), and attended monthly by many of the study participants, where housecleaners can access legal and other resources to address violations of labor rights and organize together as immigrant women. Recognizing the absence of workplace safety and health training for housecleaners, we also developed training about safer cleaning practices and approaches to discussing safer work practices with their client (17). Further development and systematic evaluation of these kinds of programs has the potential to address some of the pathways through which employment becomes a structural driver of discrimination and racial and ethnic health inequities.

### Study Limitations

Our study was limited by only including participants who were currently working and housecleaners with the most significant exposures may have been forced to leave this profession due to health issues. While our outreach efforts attempted to reach a diverse group of Latinx housecleaners and our study had an acceptable participation rate (70%), especially for such a hard-to-reach population, those who chose to participate could be a select population. Additionally, our survey items characterizing housecleaners EQ were all self-reported and we had no way to test their validity. Nonetheless, the demographic and employment characteristics of our study population are very similar to data from other national samples of housecleaners [18,19,20]. Additionally, the cross-section design of the study limits our ability to make any causal inferences and it is possible that the housecleaners’ mental or physical health status could have affected their working hours or employment conditions.

Finally, our measure of irritant work-related symptoms likely results from housecleaners’ exposures to chemical components of cleaning products, as has been shown in previous research [22,24,25], though in this analysis we did not include measures of chemical exposures. A major aim of our Safe and Just Cleaners study is to quantify the types of cleaning products used by the housecleaners in our study and measure how specific cleaning practices affect workers’ exposures. Those components of our study are ongoing and will be reported in future publications. Additionally, other potential work exposures experienced by housecleaners, such as heavy lifting, were not included in our analysis and may have also contributed to the cleaners adverse mental and physical health.

## 5. Conclusions

In this study, we explored how dimensions of employment quality together with workplace exposures can contribute to Latinx housecleaners mental and self-reported overall health. Our data characterizing the precarious working conditions for Latinx housecleaners in NYC is supported by similar findings by others. We add to this literature by demonstrating the role of poor employment quality in mental and self-reported ill-health, illustrating the pathways through which precarious work contributes to ill-health and health inequities. Our use of the EQ framework in our survey design contributed to the planning and implementation of our project’s public health educational and policy-oriented activities, a key component of our CBPR approach. Similar applications of an EQ framework could provide important insights for intervention pathways to improve health for other workers in precarious employment. Additional and more comprehensive policy approaches that address employment quality in addition to reducing workplace exposures are needed to promote greater health equity.

## Figures and Tables

**Figure 1 ijerph-19-15973-f001:**
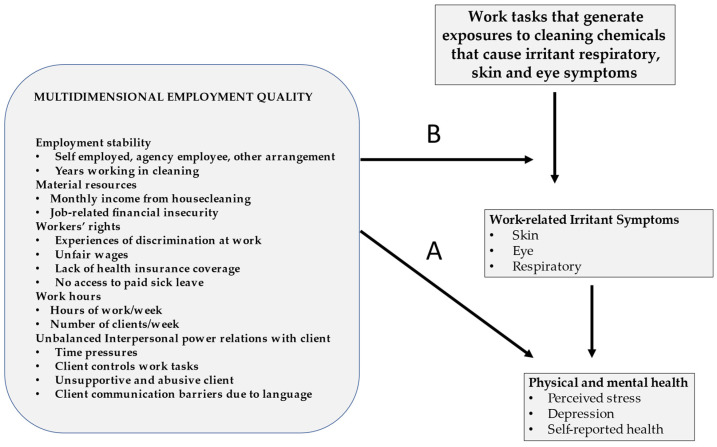
Theoretical Framework—Employment Quality (EQ), Workplace Chemical Hazards, and Health: Safe and Just Cleaners Study. Note: Pathway A hypothesizes a direct causal pathway between EQ and physical and mental health. Pathway B hypothesizes that the effects of EQ on health are mediated through work practices that generate chemical exposures causing irritant symptoms leading to poorer mental and physical health.

**Table 1 ijerph-19-15973-t001:** Participant Employment Quality Indicators: Safe and Just Cleaners Study.

	N (%)		N (%)
**Material Resources**		**Workers’ Rights**	
Monthly earning		Has no health insurance coverage	198 (49%)
0: $500 or less	149 (37%)	Unfair pay during last year	
1: >500 but <$1000	112 (28%)	Paid below the minimum wage	174 (44%)
2: >1000 but <$1500	83 (21%)	Not paid or paid less than promised	86 (22%)
3: $1500 or more	26 (6%)	Sick leave	
Prefer not to answer/missing	32 (8%)	Not provided paid sick leave	341 (85%)
Worry about having enough clients		None, paid or unpaid, without retaliation	187 (48%)
0: Never	101 (25%)	Discrimination at work due to:	
1: Sometimes	193 (48%)	Language you speak	99 (25%)
2: Almost always	75 (19%)	Being an immigrant	90 (22%)
3: Always	31 (8%)	Ethnicity	79 (20%)
Earnings sufficiency to cover expenses		Gender	41 (10%)
0: Sufficient without problem	107 (27%)	Age	37 (9%)
1: Sufficient but it was difficult	216 (54%)	Country of origin	37 (9%)
2: Insufficient	79 (19%)	Race	37 (9%)
		Sexual orientation	5 (1%)
**Employment Instability**		**Work hours**	
Years working as housecleaner in US			**Mean (SD)**
0: 6 months–<1 year	19 (5%)	Number of clients per week	3.2 (2.1)
1: 1–4 years	122 (30%)	Hours of work per week	21.7 (12.7)
2: 5–9 years	108 (27%)		
3: 10–15 years	86 (21%)		
4: >15 years	66 (16%)		
Employment arrangement			
Solo self-employed	264 (66%)		
Self-employed with others	33 (8%)		
Employed by an agency	64 (16%)		
Cleaners’ cooperative	19 (5%)		
Housecleaner day laborer	17 (4%)		

Note: Full survey sample was 402; SD = Standard deviation.

**Table 2 ijerph-19-15973-t002:** Employment Quality Indicators for Interpersonal Power Relations with Client: Safe and Just Cleaners Study.

	N (%)	N (%)	N (%)	N (%)
Interpersonal Power Relations Survey Item	Never	Sometimes	Almost Always	Always
**Time demands**				
Feel pressure to work quickly to satisfy their client	225 (57%)	93 (23%)	33 (8%)	47 (2%)
Don’t follow safety and health precautions to finish work quickly	185 (47%)	130 (33%)	49 (12%)	33 (8%)
**Client Control over work practices**				
The employer/client chooses the cleaning products they use	36 (9%)	92 (23%)	75 (19%)	199 (49%)
They decide themselves the order that they do their tasks ^a^	58 (14%)	75 (19%)	62 (15%)	207 (50%)
Their employer asks or demands they use stronger cleaning products ^b^	246 (62%)	69 (17%)	81 (20%)	
**Client social support or abuse**				
Their employer values or praises their work ^a^	19 (5%)	95 (24%)	74 (19%)	212 (53%)
Their employer provides tools needed to get your work done ^a^	10 (3%)	67 (17%)	53 (13%)	270 (67%)
They have difficulties communicating with client due to language	168 (43%)	154 (39%)	25 (6%)	48 (12%)
	**Yes**			
Experienced verbal abuse at work in the last 12 months (yes/no)	72 (18%)			

Note: Study sample was 402. ^a^ These items were reverse scored; ^b^ This item only had 3 response categories-no, sometimes, and yes.

**Table 3 ijerph-19-15973-t003:** Work-related Irritant Symptoms, Depression, Perceived Stress, and Self-Reported Health.

	N (%)	Mean (SD)
**WORK-RELATED IRRITANT SYMPTOMS**		
Skin: Rashes, itching, or redness or chapping on hands or arms that last more than one week	107 (27%)	
Eye irritation while using cleaning products		
0 = None	103 (26%)	
1 = Mild	45 (11%)	
2 = Moderate	120 (30%)	
3 = Severe	134 (33%)	
Respiratory problems that improve when away from work:		
Health care provider diagnosed asthma	20 (5%)	
Nighttime shortness of breath	54 (13%)	
Either asthma diagnosis and/or nightime shortness of breath	65 (16%)	
**PHYSICAL AND MENTAL HEALTH OUTCOMES**		
Depression (CESD-10)		6.19 (5.34)
Score ≥ 10	93 (24%)	
Cohen’s perceived stress		12.68 (6.44)
Moderate/severe score ≥ 14	173 (44%)	
Self-reported health		
Excellent	31 (8%)	
Very Good	71 (18%)	
Good	182 (45%)	
Fair	111 (27%)	
Poor	7 (2%)	

Note: Full survey sample was 402; SD = Standard deviation.

**Table 4 ijerph-19-15973-t004:** Principal Component Analysis (PCA) Results for Employment Quality (EQ) and work-related irritant symptoms survey questions that could be combined into a single variable: Safe and Just Cleaners Study.

Employment Quality Measure	N	Score Range	Mean (SD)	Component Loading
**Interpersonal Power Relations with Client**				
Time pressures at work ^a^	393	0–6	1.57 (1.60)	
Feels pressure to work quickly to satisfy their client	397	0–3	0.75 (1.03)	0.84
Doesn’t follow safety and health precautions to finish work more quickly	397	0–3	0.82 (0.95)	0.71
Client controls work tasks ^a^	396	0–8	3.63 (2.97)	
Employer chooses the cleaning products they use	402	0–3	2.09 (1.04)	0.53
Cleaner does not decide themselves the order of their cleaning tasks	402	0–3	0.96 (1.13)	0.68
Employer asks/demands they use stronger cleaning products	396	0–2	0.58 (0.81)	0.77
Unsupportive and verbally abusive client ^a^	397	0–7	1.51 (1.60)	
Employer doesn’t value or praise your work	400	0–3	0.80 (0.96)	0.86
Employer doesn’t provide the tools needed	400	0–3	0.54 (0.86)	0.66
Experienced verbal abuse at work in the last 12 months	401	0–1	0.17 ^b^	0.61
**Material Resources**				
Job-related financial insecurity ^a^	400	0–5	2.02 (1.29)	
Earning sufficiency for basic needs during the last year	402	0–2	0.93 (0.68)	0.86
Worry for having enough clients to earn the money they need	400	0–3	1.09 (0.86)	0.86
**Workers’ Rights**				
Unfair wages in last year ^a^	393	0–2	0.66 (0.75)	
Paid below the minimum wage, in the past year	398	0–1	0.44 ^b^	0.90
Paid less than what your employer agreed	397	0–1	0.22 ^b^	0.78
No access to sick leave ^a^	392	0–2	1.32 (1.30)	
Not provided sick leave	401	0–1	0.85 ^b^	0.90
Cannot take time off, paid or unpaid, without retaliation	392	0–1	0.47 ^b^	0.74
**Irritant symptoms consistent with work-related chemical exposures**				
Work-related irritant symptoms ^a^	402	0–5	2.14 (1.50)	
Work-related skin rash	402	0–1	0.27 ^b^	0.80
Work-related respiratory symptoms	402	0–1	0.16 ^b^	0.80
Eye irritation when cleaning	402	0–3	1.72 (1.18)	0.70

Note: Total survey sample = 402; SD = Standard deviation; ^a^ Variable name given to the combined variable ^b^ For items with yes/no response options the mean represents the frequency of yes responses and no standard deviations were calculated.

**Table 5 ijerph-19-15973-t005:** Multiple linear regression model for associations between employment quality dimensions and work-related irritant symptom score.

Model Fit	R^2^ = 0.25 *p* < 0.01
	Beta (*p* value)
**Worker rights**	
Number of experiences of discrimination at work	0.14 (*p* < 0.01)
Not having health insurance	0.25 (*p* = 0.05)
**Material resources**	
Job-related financial insecurity	0.21 (*p* < 0.01)
**Interpersonal power relations with client**	
Time pressures	0.10 (*p* = 0.05)
Unsupportive and abusive client	0.20 (*p* < 0.01)

**Table 6 ijerph-19-15973-t006:** Multivariable Logistic Regression Models for Depression, Perceived Stress and Self-Reported Health: Safe and Just Cleaners Study.

	STRESS (PSS ≥ 14)		DEPRESSION (CESD-10 ≥ 10)		FAIR/POOR HEALTH	
	Model 1	Model 2	Model 1	Model 2	Model 1	Model 2
	OR (95% CI)	OR (95% CI)	OR (95% CI)	OR (95% CI)	OR (95% CI)	OR (95% CI)
**INDIVIDUAL FACTORS**						
Comfortable with English ^a^	–	–	–	–	0.58 (0.38–0.88)	0.54 (0.35–0.87)
Primary wage earner	–	–	2.07 (1.17–3.64)	2.05 (1.14–3.67)	–	–
**EMPLOYMENT QUALITY**						
**Employment stability**						
Years of work as cleaner ^b^	–	–	–	–	1.53 (1.20–1.96)	1.58 (1.23–2.04)
Worked for an agency	–	–	0.40 (0.18–0.88)	0.36 (0.16–0.80)	–	–
**Work hours**						
Hours worked per week (per additional hour)	–	–	–	–	1.03 (1.00–1.05)	1.02 (1.00–1.05)
**Interpersonal client relations with the client(s)**						
Time pressures ^c^	–	–	–	–	1.34 (1.12–1.61)	1.27 (1.06–1.52)
Unsupportive/abusive ^d^	1.17 (1.01–1.36)	1.09 (0.93–1.27)	1.33 (1.13–1.57)	1.23 (1.03–1.47)	–	–
Language-related communication barrier ^e^	1.61 (1.25–2.08)	1.61 (1.25–2.08)	–	–	–	–
**Workers’ rights**						
Unfair wages ^f^	1.41 (1.02–1.93)	1.33 (0.96–1.84)	1.81 (1.27–2.58)	1.72 (1.19–2.48)	–	–
Experiences of discrimination at work ^g^	–	–	–	–	1.21 (1.02–1.43)	1.10 (0.92–1.31)
**Material resources**						
Job-related financial insecurity ^h^	1.46 (1.20–1.76)	1.36 (1.11–1.65)	1.78 (1.41–2.22)	1.63 (1.29–2.06)	–	–
Higher monthly earnings ^i^	–	–	–	–	0.72 (0.55–0.96)	0.72 (0.54–0.96)
**WORK-RELATED IRRITANT SYMPTOMS ^j^**	Not tested	1.37 (1.14–1.63)	Not tested	1.54 (1.24–1.91)	Not tested	1.51 (1.22–1.88)

Note: PSS ≥ 14 = Cohen’s Perceived stress scale score above 13; (CESD-10 ≥ 10) = Center for Epidemiologic Studies Depression 10-item Scale above 9; Poor/Fair Health versus Good, Very Good or Excellent on the self-reported health item; OR = Odds Ratio; 95% CI = 95% confidence intervals. Variables not included in a final model because of lack of statistical significance are indicated with a –. ^a^ 3 pt response score Ref = uncomfortable; ^b^ 5 pt response score Ref ≤ 1 yr and each point adds 5 years; ^c^ 7 pt response score Ref = no time pressure; ^d^ 8 pt response score Ref = high support/no verbal abuse; ^e^ 4 pt response score Ref = no communication barriers; ^f^ 3 pt response score Ref = fair wages; ^g^ 9 pt response score Ref = no experience of discrimination; ^h^ 6 pt response score Ref = sufficient earning/job security; ^i^ 4 pt response score Ref ≤ $500/months and each point adds $500 in monthly earning; ^j^ 6 pt response score REF = no irritant symptoms.

## Data Availability

The data presented in this study are available on request from the corresponding author. The data are not publicly available in order to protect the confidentiality of the participants.

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
