# Peer review of "Employment Quality and Mental and Self-Reported Health Inequities among Latinx Housecleaners: The Safe and Just Cleaners Study"

_ijerph, 2022, doi:10.3390/ijerph192315973_

Round 1

Reviewer 1 Report (New Reviewer)

The paper is an important exploration of the health and wellbeing of Latinx Housecleaners in and around New York City. The survey method is thorough and comprehensive, and illustrates both internal and external validity. However, the study is (more than once) referred to as CBPR and it is not represented as such. A true CBPR study would engage the community and ask the questions *they* find important, which is not how this research is portrayed. If that was the process, it needs to be made more clear. The authors did discuss focus groups and interviews, but this was to narrow the questions, not the topic and interest. In CBPR, the basic question and area of concern would come from the community. Instead, this seems more like a correlational study using an expansive survey and rigorous statistical analysis. This paper would be stronger framing it as such. Regardless, the methods should be clarified. 

Author Response

Reviewer 1:

  1. The paper is an important exploration of the health and wellbeing of Latinx Housecleaners in and around New York City. The survey method is thorough and comprehensive, and illustrates both internal and external validity. However, the study is (more than once) referred to as CBPR and it is not represented as such. A true CBPR study would engage the community and ask the questions *they* find important, which is not how this research is portrayed. If that was the process, it needs to be made more clear. The authors did discuss focus groups and interviews, but this was to narrow the questions, not the topic and interest. In CBPR, the basic question and area of concern would come from the community. Instead, this seems more like a correlational study using an expansive survey and rigorous statistical analysis. This paper would be stronger framing it as such. Regardless, the methods should be clarified. 

Thank-you for the positive feedback.  Regarding your comment about the CBPR process, we had not initially put a lot of detail about very extensive CBPR process in the manuscript. Our project included many  activities in addition to the survey that exemplify our commitment to the CBPR process  (see our website www.safeandjustcleaners.org ). The goal of this manuscript was to share our survey results through the peer reviewed literature.  Other articles and presentations focus more exclusively on our process. However, the point you raise is important and we agree that it is important to always explain the process beyond just calling it CBPR. We have added a new section at the start of the methods that includes an expanded overview of our CBPR process. Our research question arose out of a previous multi-year collaboration with the partner community-based organization.  While the overall research project was focused on chemical exposures, the issue of employment quality was emphasized by the community and our development of employment quality items was a result of their concern over these issues. Our section in the discussion which describes how findings have been used as part of our action project components also reflects our commitment to research supporting community action.

The initial paragraph of the methods section that has been added reads as follows:

Our overall study used a CBPR approach that emphasized the following elements of community involvement: power and control, responsibility and ownership, participation, and influence [26]. This academic-community partnership arose out of a previous collaboration with our community partner, Make the Road New York (MRNY), to create occupational health training for Latinx construction laborers and cleanup workers following a natural disaster in New York City [27]. To further develop our mutual interest in workplace exposures to cleaning chemicals and to expand our research to include more women workers, together our community-academic partnership proposed and carried out this study of household cleaners. Many of the dimensions of employment quality included in our study were initially raised, developed, and prioritized by our community partner. Our overall project activities attempted to balance research and action for the mutual benefit of all partners, another key CBPR principle.

Reviewer 2 Report (New Reviewer)

Thank you for an interesting read. I think that my contribution for this peer review is mainly that of a scholar of cultural sciences. My observations concern the positioning of the study in relation to the previous research and in the wider global framework.

Rows 42-46 I would be interested hearing where the previous studies that are similar to this one have been conducted, in the US or elsewhere, and how this current study is contributing to the global scale research efforts.

Row 60-61 inequalities, where? in the US? or around the world?

Row 70-71 maybe would be interesting to know where EQ has been implemented before, in which contexts

Row 76 where there any challenges regarding the use of two languages in the research process

Row 79 why was New York chosen as the research context

Was something surprising in the research results? Did you come up with ideas what should be studied more? Why this focus group is particularly important/interesting for research?

Author Response

Reviewer 2:

Thank you for your comments and we appreciate the perspective of a scholar of cultural sciences.

  1. Rows 42-46 I would be interested hearing where the previous studies that are similar to this one have been conducted, in the US or elsewhere, and how this current study is contributing to the global scale research efforts.

Most previous studies have been done in the United States and Europe and this has been added in that section.  The contribution of this study to the body of literature is addressed in the conclusions section as described in point #6 below.

  1. Row 60-61 inequalities, where? in the US? or around the world?

We have clarified that this statement and the statistics it draws on refers to the United States

  1. Row 70-71 maybe would be interesting to know where EQ has been implemented before, in which contexts

We have clarified that these studies have mainly been conducted in the United States and Europe

  1. Row 76 were there any challenges regarding the use of two languages in the research process

We have clarified that the survey was done in Spanish. We have also added a few more details regarding the process of drawing on some English language survey items but adapting them for use in Spanish.  That section now reads as follows:

We initially developed or identified survey items in English and then bilingual team members translated them into Spanish, except when using previously validated translations of existing measures. When selecting items in English the study team, considered the items’ cultural or linguistic translatability for a NYC-based Spanish speaking population. In addition, we performed cognitive and pilot testing to assure that survey items were interpreted as intended and made edits directly to the Spanish version, when indicated, prior to fielding the survey [41].

  1. Row 79 why was New York chosen as the research context

This is now covered in the added section describing our community-based participatory process (see the first paragraph of the methods section).  Our research team and community partners, who we had been collaborating with previously, were all based in NYC.

  1. Was something surprising in the research results? Did you come up with ideas what should be studied more? Why this focus group is particularly important/interesting for research?

We feel that we have addressed what we felt were the most significant findings and suggestions for additional research in the discussion, but this is specifically addressed in our conclusions section as highlighted below.  We are not sure what the reviewer is referring to by “focus group” since this research involved much more comprehensive data collection beyond just focus groups.

5.0 Conclusions

In this study, we explored how dimensions of employment quality together with workplace exposures can contribute to Latinx housecleaners mental and self-reported overall health. Our data characterizing the precarious working conditions for Latinx housecleaners in NYC is supported by similar findings by others. We add to this literature by demonstrating the role of poor employment quality in mental and self-reported ill-health, illustrating the pathways through which precarious work contributes to ill-health and health inequities. Our use of the EQ framework in our survey design contributed to the planning and implementation of our project’s public health educational and policy-oriented activities, a key component of our CBPR approach. Similar applications of an EQ framework could provide important insights for intervention pathways to improve health for other workers in precarious employment. Additional and more comprehensive policy approaches that address employment quality in addition to reducing workplace exposures are needed to promote greater health equity.

This manuscript is a resubmission of an earlier submission. The following is a list of the peer review reports and author responses from that submission.

Round 1

Reviewer 1 Report

Referee report on ijerph-1799846
General summary of the paper
The authors did an interesting work to study the impact of employment quality on
depression, perceived stress, and self-reported ill-health. The study is great since the
authors focused their attention on a specific group of workers, i.e., house-cleaners working
in urban apartments and suburban homes in New York City.
I must clarify that my expertise is low regarding the type of items used to describe
the employment quality and the aspects of household cleaning work, so my comments
are limited to statistical aspects. In my opinion, the study design is quite adequate, and
the statistical methods are correctly presented in general, some parts, however, need to
be clarified and/or justified.
1 Materials and Methods
1. Line 219: The authors wrote “Once the summary EQ and irritant symptom measures
were developed, we examined predictors of the measures using multiple linear
regression.” It is not clear the data analysis processes. If I understand correctly,
the authors made a multiple linear regression using as dependent variables the
relevant items found by PCA. Please specify. Therefore, if it is the case, why did
the authors use a multiple linear regression having as dependent variables (in some
cases) an ordinal categorical variable? Please justify it, and It could be helpful to
see some diagnostic plots as supplementary materials.
2. Line 222: The authors wrote: “The measure of the predictive value of the linear
regression models was based on the significance levels associated with each of the
predictors and the R-Square provided the Goodness of Fit measure”. However,
the R squared index is not a measure of predictive ability. Also, I think the authors
fitted models to illustrate the relationship between covariates and dependent
variables rather than predict them. In addition, I would suggest adding the effect
size for each statistical test performed to measure the strength of the relationship
tested.
3. Line 225: The authors wrote: “using standard threshold cut points for each of
the outcome measures”. Please specify the value of the standard threshold.
1
2 Results
1. Line 287: The authors said that PCA reduced the EQ items from 22 to 14 (8
single item and 6 multi-item). But looking at Table 3 I can see 7 single item and
7 multi-item.
2. Line 287: I like the PCA application, but it is difficult to see the results in Table
3, I would suggest adding some plots for the most relevant dimensions.
3. Line 314: I don’t get why in Table 5 the Irritant work-related Symptoms are
measured by a scale from 0 to 6, but in Table 4 the variables is described by three
items. I imagine that the authors combined these items. Please clarify.
4. Table 5: Please also report the beta parameter estimates and their corresponding
p-values even if it is not significant, and underlines (with a star or something else)
the significant ones. If the covariates do not enter in the model please put a line
or something else, and specify it in the caption.
5. Table 5: The authors performed seven different models using the seven dependent
variables describing the IrritantWork-related Symptoms, and Employment Quality
Multi-Item Measures. Therefore, they perform seven statistical tests for the same
covariate, e.g., Comfortable with English. Did the authors adjust for multiple
testing? I recall here one of my previous suggestions, i.e., to insert also the effect
size.
6. Table 5: The authors performed one statistical model for each Employment Quality
Measure (14), andWork-related irritant symptoms (3). However, Table 5 shows
only seven models; what happens? Please justify properly. I assumed they selected
the models considering the significance of the statistical tests for the R2 index. If
it is the case, please insert the results also if the R2 is not significant. I would see
some diagnostic plots, since this index depends on the variance of the independent
variables and the variance of the residuals.
3 Minor
1. Please describe properly the tables’s captions.

Reviewer 2 Report

Review of the manuscript:

Employment Quality and Mental and Self-Reported Health Inequities Among Latinx Housecleaners: The Safe and Just Cleaners Study

The manuscript deals with a relevant research question of association between work quality and health. The data were collected on a sample (N=402) of precarious workers – immigrant Latinx housecleaners in the New York City metropolitan area. The authors conducted the qualitative part of the research (i.e. focus groups) with the aim of constructing a survey. The survey applied in the quantitative part of the research was quite long – it lasted 60-75 minutes. General hypotheses on the connection between the quality of employment and health are presented, and a model is presented that defines direct and indirect connections between sets of variables. A number of statistical analyzes were conducted, and the research findings are based on descriptive statistics, correlations between research variables and linear regression analysis.

Although the research collected important data on workers who are underrepresented in the literature, the findings and conclusions obtained are not clear primarily due to the unclear strategy of statistical analysis of these data.

I suggest major revisions of the chapters "2.0. Materials and Methods" and "3.0. Results" before the manuscript is resubmitted as a scientific paper. In addition, clear research hypotheses should precede these chapters. Here I present the main objections and requests:

1. The hypothesis and following elaboration: “We hypothesized that the combination of poor EQ and hazardous exposures together contribute to housecleaners’ health and well-being. …”   (p.2, line 87-98) is general, and research aims are not clear. The model presented in the Figure 1 (p.4) is pretty complex and it is not clear how it was tested in the study.

Precise research objectives should be defined as well as research hypotheses that define the expected relationships between the research variables.

2. The purpose and results of the qualitative study are not clear: How did it help define the survey? What were the qualitative results? The description of the applied survey is not structured, and it is very complicated (impossible for me) to follow the text. The applied scales should be described in separate paragraphs with data on the items number and content, the method of answering, metric characteristics if applicable (internal consistency or reliability).

3. The data analysis strategy and the performed data analysis per se are not at all clear. In addition, the reported results are too extensive and incoherent. The results of the regression analyses are not clear, just as it is not clear how the mediations were tested.

I suggest omitting FA and basing results on clear and correctly presented descriptive statistics – M, SD, Range, test of normality of distribution on cumulative measures whenever possible (homogeneity of scales); bivariate correlations and linear regression analysis. I suggest the authors consult the literature in the field (eg Hayes, A.F. (2022). Introduction to Mediation, Moderation, and Conditional Process Analysis (3rd ed)).

Reviewer 3 Report

Reviewers Comments:

This is an interesting paper on a well-designed study. The study emphasizes the need for greater attention on employment as a determinant of health and health outcomes in the immigrant and Latinx population. Findings from the research also provide a framework for future public health and community-based research. This includes developing and using models, such as the employment quality framework, to understand better the association between the determinants of health and health outcomes in the underserved and minority populations. We make several suggestions below that should improve readability and understanding overall; we also point out our challenge in understanding the methodology and results associated with the linear regression analysis, which we feel would benefit from a significant revision. Please note that, as a learning experience for a doctoral student, there were two reviewers for this article, hence the use of “we” throughout the review (and not an attempt to infer royalty status).

1.     Title and Abstract

a.     The manuscript’s title and abstract make no mention of the cross-sectional study design. This could be included in the abstract, and potentially also the title.  

b.     The abstract makes no mention of the project timeline. The timeline, as stated in line 89 can be added to the abstract.

c.      Last, study results described in the abstract does not include quantified associations (Lines 22-29). Please consider adding the numeric odds ratios, along with confidence intervals.

2.     Introduction

a.     The hypothesis (lines 87-88) is vaguely worded. If, as found in other studies, there is a correlation between poor EQ and health, you could state this directly rather than just implying that there is some association.

b.     Figure 1 is particularly helpful in understanding the study purpose and the hypotheses. Please consider referring to this figure in the introduction (even earlier than it is currently referenced). For example, it could be referenced in line 90: “…and the association of those EQ measures with ill-health (see Figure 1)”.

3.     Methods

The suggestions fall into two main categories, with detailed notes under each heading.

Improving readability

a.     Lines 200-201: the percentages (92% and 8%) should not be in parentheses, as they are the number being reported, i.e.,  “…of those eliminated, 92% had not worked…” UNLESS you report the actual number and then include the percentage in parentheses. Readability will also be improved if lines 201-202 follow the same convention; for what it’s worth, we prefer the number (percentage) format. You might also consider using a flow diagram to show where participants were excluded from the analysis and showing the different percentages (including the variation by outreach approach, i.e. 62-75%).

b.     Lines 210-218: The authors appear to have used PCA to develop individual dimension scores for each of the 5 EQ domains, but a single irritant domain (as opposed to 3 domains). However, the wording on this was a bit unclear, since “single score” is used in both cases and could mean 1 total domain or 5 for EQ and 1 total domain or 3 for irritant. Also, “score” and “scale” seemed to be used somewhat interchangeably, leaving the reader to wonder if these are synonyms or should be thought of separately. If the point of PCA was merely to reduce the number of items, but then complete the remaining analyses using single items, the explanation should reflect that rather than trying to estimate single domain scores.

Clarifying methods

a.     Lines 202-203: The authors state “402 had data complete enough to be included in the analysis”, but what constitutes being complete enough is not defined. Please add this information.

b.     Lines 219-222: The authors refer to summary EQ and irritant symptom measures as outcomes to be modeled, which would follow from the previous paragraph to mean single-domain scores. Yet, Table 5 reports information for the individual items within the domains. This makes the interpretation of Table 5 difficult and does not match with the PCA analysis to create single-domain scores, but instead looks like every item was evaluated as a predictor of every other item. If this was the point, it is not entirely clear why a simple correlation matrix would not be sufficient. Additionally, no information is provided in methods to inform the interpretation of Table 5; for example, when no coefficient and p-value appear, is that because this was dropped from the model (and was that a decision made by the authors or was it a machine-based decision, such as backward elimination, which was discussed for the logistic regression model) or for some other reason? The assumption we made in reviewing Table 5 was that independent variables were treated as categorical, but a statement to that effect in the methods would be helpful.

c.      Line 227: The authors state “…using standard threshold cut points…” Please add a citation(s) for these standards. It may also be helpful to indicate the value of the cut-point(s) for the outcome(s).

d.     Lines 228-230: Does “…valid answers for 9 of 10 measures…” mean that the participant supplied an answer (i.e., answer completion) or was there some method for assessing whether or not an answer was valid. This needs clarification, and potentially a citation if there is a methodology related to validity. Is this part of the methodology for these scales (i.e., published) or determined by the authors?  Also, was this applied to other measures besides CESD-10 and PSS, and if so, how?  

e.     Line 231: Non-SAS users may be unfamiliar with the Max Rescaled R-Square measure. We suggest indicating you used the adjusted R-square measure as programmed for SAS, and provide a citation (if you don’t already have one, this might be sufficient: https://support.sas.com/resources/papers/proceedings/proceedings/sugi25/25/st/25p256.pdf).

f.       Lines 233-235: The authors state “…if the addition of irritant symptom scale reduced the significance of one or more of the EQ predictor measuresit was assumed that the irritant symptom scale at least partially mediated the effect of that EQ measure…” Was there a percent change required, or some other metric for determining what was an important change? Is there a citation for this methodology, or did the authors determine the amount? Defining what is an important amount of change may also impact your results narrative (lines 355-363). 

4.     Results

a.     Lines 273-283: Please check the percentages described in text with those in Table 2. For example, in line 276; “36% always/almost always decided order of cleaning tasks….” But Table 2 shows 65% for the sum of these two categories (p.8). Also, in lines 282 and 283, “Many clients (53%) reported communication barriers, at least sometimes,….” but in Table 2, the sum of these three categories is 57%.

b.     Lines 294-299 & Table 4: The authors state that the “…the items were combined as a multi-item 6-poitn scale.” (line 298), but there are 8 items in Table 4. The assumption we made when reading this was that the “asthma” and “shortness of breath” rows in the table are sub-categories of “any respiratory problems and were not included in the item counting. If this is the case, we suggest sub-setting the numbers to the right in the table, and clearly stating how the scale was constructed in the narrative.

c.      Line 234: The Irritant Symptoms Scale in line 234 needs more description. There is reference to determining whether the various items can be grouped into a single score (lines 217-218), but no description given of what the grouping may mean or how it was entered into the model (line 217-8).

d.     Lines 299-303: Please begin a new paragraph for the sentence “Using standard cut-points…”, as the previous lines are about the irritant chemical scale. Please provide citations for the standard cut-points for CES-D-10 and Cohen’s PSS (line 300). Please start a new sentence for self-reported health, as this doesn’t have cut-points other than the categories available for selection by the respondent.

e.     As noted in Methods (item b), the interpretation of Table 5 needs clarification. Based on the modifications made to the methods and to Table 5, section 3.6 may need to also be re-written. Lines 307-313 reference association, which suggests something like a simple correlation matrix, while lines 314-325 reference multiple linear regression. Yet both sections refer to Table 5, and beta coefficients are provided, implying this is the regression output. All in all, we were not sure what to make of section 3.6 and Table 5, and the related methodology about the linear regression analysis.

f.       Lines: 337-339: You suggest there is evidence of dose-response for hours worked per week and fair/poor health, yet the confidence interval includes 1 in both models. How broad is your range of hours of work for the dose? Is this finding more likely suggestive of a relationship and needs to be further explored in additional studies?

g.     Lines 341-342: You might consider referencing Table 5 as the place where you report on “…agency work was associated with higher levels of unbalanced power relations…”

5.     Discussion

a.     Lines 395-404: Are you basing your conclusions in these sentences on some measure of immigration status in your analysis (which has not been reported in the manuscript) or on the work of others? These sentences imply that some of this is from your findings, but we did not see any findings reported by immigration status.

b.     Line 414: Is this a typo and should be “Make the Road New York” (rather than Yor)?

c.      Line 422: Is this a typo and should be “collection of survey data” (rather than collection our survey data)?

d.     Statement made in lines 421 and 422 (“…given that the COVID pandemic began shortly after we completed collection…”), is a little misleading. The study period is listed in line 89 as August 2019 and February 2020. New York was one of many cities already experiencing high infection rates between December and February 2020. The statement may need to be reworded.

e.     Information for implications for future research might include potential generalizability to other minority populations not included in lines 405 (actionable findings) or 454 (conclusions). Further, we recommend a statement that encompasses the idea that other studies could consider using this (or similar) framework to address other determinants of health including housing, education, or health care, in other underserved communities like the low-income and racial/ethnic minorities. 

f.       Additional information also needed in line 439 (limitations). No information provided regarding the impact of COVID-19 on increased pressure at work. The rising rates of COVID-19 during the study period may help explain observed client’s behavior towards housecleaners, such as choice of cleaning products and order of cleaning tasks.

Reviewer 4 Report

The manuscript is written in a concise and pleasant way. The authors created a questionnaire for the analysis of EQ on the basis of an accurate analysis of the occupational conditions of the category. The sampling was carried out with scrupulous methodology. The research provides interesting results from a social and economic point of view as well as a health one.

One minimal modification that the authors could introduce concerns the limitations. The authors chose PSS to measure stress, a questionnaire that evaluates life stress rather than occupational stress. Considering that this study investigates occupational hazards, it would have been more useful to use a specific questionnaire for occupational risk, for example Siegrist's effort / reward imbalance which is also made up of 10 questions. This may be a limitation of the study that the authors could discuss if they find it useful.

Author Response

Response to Reviewer 4

The manuscript is written in a concise and pleasant way. The authors created a questionnaire for the analysis of EQ on the basis of an accurate analysis of the occupational conditions of the category. The sampling was carried out with scrupulous methodology. The research provides interesting results from a social and economic point of view as well as a health one.

One minimal modification that the authors could introduce concerns the limitations. The authors chose PSS to measure stress, a questionnaire that evaluates life stress rather than occupational stress. Considering that this study investigates occupational hazards, it would have been more useful to use a specific questionnaire for occupational risk, for example Siegrist's effort / reward imbalance which is also made up of 10 questions. This may be a limitation of the study that the authors could discuss if they find it useful.

Thanks for your very supportive comments. We appreciate the positive feedback.

In response to your suggestion regarding using a questionnaire more focused on occupational stress. We did measure indicators of job stress given that many of the EQ measures overlap with job stress measures.  Our interpersonal client power relations survey items were drawn from the Job-content model which is one very common way of measuring job stress similar to the Seigrist effort/reward imbalance. Generally these job related stressors are used as independent variables with outcome variables like those we used in our study as well as other health outcomes like work-related injuries and cardiovascular disease  We have tried to clarify this by expanding the section starting on line 171 which now reads:

“In developing these survey items, we drew on the well-developed job content construct used to characterize psychosocial work environment and job stress[32], since in our focus groups housecleaners emphasized the importance of time demands, limitations on housecleaners’ power to make decisions about cleaning product use and practices, and the degree to which their clients were supportive or abusive.”

Reviewer 5 Report

A very good paper; I enjoyed reading it.  There is simply nothing wrong with it.  Worth publishing as soon as the only two minor mistakes are corrected

Review report

“Employment Quality and Mental and Self-Reported Health Inequities Among Latinx Housecleaners: The Safe and Just Cleaners Study”

There are simply not enough studies looking into the physical and mental health of vulnerable segments of the population, as this research does brilliantly.  So, the subject of this paper is important and relevant.  Furthermore, everything about this paper is absolutely adequate: review of the literature, methodology, presentation and discussion of the results, etc.  The variety of outreach methods used is a clear indication of the sensitivity of the researchers to the precarious status of their study population.  A picture, they say, is worth a thousand words.  The theoretical framework in Figure 1 is a clear example of that saying.  It is so well done that one understands the entire research at a glance.

I particularly liked their section “Actionable findings”; here is a research that does more than just “looking into things” or “describing a phenomenon”; having identified the poor working and health conditions of Latinx housecleaners, actions are already under way, and I am impressed.

There are hardly any mistakes in this paper; the only two I found are:

·       Line 252: Tables (plural)

·       Line 414: New York (missing letter)

In view of the comments above, my recommendation is quite simple: publish this paper as soon as the two typos above are corrected.

Author Response

Response to Reviewer 5

There are simply not enough studies looking into the physical and mental health of vulnerable segments of the population, as this research does brilliantly.  So, the subject of this paper is important and relevant.  Furthermore, everything about this paper is absolutely adequate: review of the literature, methodology, presentation and discussion of the results, etc.  The variety of outreach methods used is a clear indication of the sensitivity of the researchers to the precarious status of their study population.  A picture, they say, is worth a thousand words.  The theoretical framework in Figure 1 is a clear example of that saying.  It is so well done that one understands the entire research at a glance.

 Thanks-you very much for the positive feedback

I particularly liked their section “Actionable findings”; here is a research that does more than just “looking into things” or “describing a phenomenon”; having identified the poor working and health conditions of Latinx housecleaners, actions are already under way, and I am impressed.

Thank you for the positive feedback.  Our CBPR partnership is very committed to translating our findings into positive actions that will improve working conditions

There are hardly any mistakes in this paper; the only two I found are:

  • Line 252: Tables (plural)
  • Line 414: New York(missing letter)

 We have made these two typographical changes

In view of the comments above, my recommendation is quite simple: publish this paper as soon as the two typos above are corrected.